# Complicated Structure Change during Capillary Extrusion of Binary Blends of Polycarbonate and Poly(methyl methacrylate)

**DOI:** 10.3390/ma15082783

**Published:** 2022-04-10

**Authors:** Masayuki Yamaguchi, Kodai Nakamura, Takeyoshi Kimura, Nantina Moonprasith, Takumitsu Kida, Kyoko Tsubouchi, Takaaki Narita, Tatsuhiro Hiraoka

**Affiliations:** 1Japan Advanced Institute of Science and Technology, School of Materials Science, 1-1 Asahidai, Nomi 923-1292, Ishikawa, Japan; s2010144@jaist.ac.jp (K.N.); s2120012@jaist.ac.jp (T.K.); moonprasith.n@jaist.ac.jp (N.M.); tkida@jaist.ac.jp (T.K.); 2Sirindhorn International Institute of Technology, Thammasat University, 99 Moo 18, Paholyothin, Khlong Luang, Pathum Thani 12120, Thailand; 3Hiroshima R&D Center, Mitsubishi Chemical Corporation, 20-1 Miyukicho, Otake 739-0693, Hiroshima, Japan; tsubouchi.kyouko.ma@m-chemical.co.jp (K.T.); narita.takaaki.ma@m-chemical.co.jp (T.N.); hiraoka.tatsuhiro.ma@m-chemical.co.jp (T.H.)

**Keywords:** polymer blends, pressure, shear, miscibility, polycarbonate, poly(methyl methacrylate)

## Abstract

The effects of pressure and shear rate on the miscibility of binary blends comprising bisphenol-A polycarbonate (PC) and low molecular weight poly(methyl methacrylate) (PMMA) were investigated using a capillary rheometer. Both pressure and shear rate affected the miscibility. The examination of an extruded strand of the blend provided information about the cause of the phase change. Under high pressure, pressure-induced demixing occurred at temperatures below the lower critical solution temperature (LCST) of the blend. Consequently, the extruded strand became opaque throughout. During shear-induced mixing/demixing, a part of the strand became opaque because of the distribution of the shear rate in the strand. For example, during shear-induced demixing, only the exterior of the strand, i.e., the high shear rate region, became opaque. Above the LCST, shear-induced mixing occurred, and only the center region of the strand became opaque.

## 1. Introduction

There has been extensive research into blends of bisphenol-A polycarbonate (PC) and poly(methyl methacrylate) (PMMA) because they are both important transparent plastics. They are immiscible at processing temperatures. Consequently, blends of PC and PMMA have phase-separated structures that cause intense light scattering. The blends have a phase diagram with a lower critical solution temperature (LCST) [1,2,3,4] and a small positive value of the Flory–Huggins parameter, e.g., 0.039 ± 0.004 at 250 °C [1]. Therefore, when one of the polymers has a low molecular weight, the system becomes miscible [5,6]. The addition of PMMA to PC without loss of transparency is desirable because it improves the surface hardness and scratch resistance of the blend [5,7,8]. However, the effects of shear flow and pressure on the miscibility of such blends have not been completely elucidated, even though conventional processing operations are performed in flow fields under pressure. For example, injection molding is sometimes performed at very high pressures, e.g., 200 MPa [9,10], and high shear rates, owing to advances in the design of processing machines. In particular, plastics with high glass transition temperatures, such as PC, are processed at a high pressure and high shear stress. In general, viscosity enhancement by the application of pressure becomes obvious beyond 10 MPa owing to the decrease in the free volume fraction [11]. At 100 MPa, the viscosity increases by 7.4 times [11]. The decrease in the free volume fraction is also responsible for the increase in the glass transition temperature *T_g_* [12].

As discussed previously, the Flory–Huggins parameter is composed of two contributions, i.e., an interaction contribution and a free volume contribution [13]. A decrease in the free volume indicates that the interaction contribution has become important. Therefore, the pressure applied during injection molding has a considerable potential to affect the miscibility of a blend. In fact, there have been reports on pressure-induced mixing/demixing [14,15,16].

Theoretically, exposure to shear flow also has the potential to affect miscibility, as discussed by several researchers [17,18,19,20,21,22]. In the present study, the effects of pressure and shear rate on the miscibility of blends of PC and PMMA were studied using a pressure-driven capillary rheometer. Because the shear rate is a function of the distance from the center of the extruded strand during pressure-driven shear flow, the miscibility, and thus the morphology, must vary according to the position once the shear-induced phase change occurs. Furthermore, the impact of pressure can be evaluated in the flow field.

## 2. Materials and Methods

### 2.1. Materials

A commercially available PC (Iupilon S2000; Mitsubishi Engineering Plastics, Tokyo, Japan) and a low molecular weight PMMA were used. The number- and weight-average molecular weights of the PC, evaluated by size exclusion chromatography (HLC-8020; Tosoh, Tokyo, Japan) with polystyrene as the standard, were *M_n_* = 2.8 × 10^4^ and *M_w_* = 5.7 × 10^4^. The *M_w_* of the PMMA was 1.3 × 10^4^ with poly(methyl methacrylate) as the standard. The *M_w_* value is almost the same with the entanglement molecular weight of PMMA, i.e., ca. 1.0 × 10^4^ [23]. Furthermore, the details of the rheological properties for the PC sample were described elsewhere [24].

### 2.2. Sample Preparation

After vacuum drying at 80 °C for 4 h, melt-blending was performed using an internal mixer (Labo Plastmill 10M100; Toyo Seiki Seisakusho, Tokyo, Japan). The blade rotation speed was 30 rpm, which provided a shear rate of 29 s^−1^ between the blades and the inner wall. The temperature was 250 °C following previous researches on PC/PMMA blends [5,6]. Mixing was carried out for 5 min. The PMMA contents were 20 and 30 wt%. Films (500 µm thick) were prepared using a compression molding machine at various temperatures before cooling to 25 °C. The sample preparation methods were the same as those used in previous researches [5,6,24].

### 2.3. Measurements

The transparencies of the compression-molded films with 500 µm thickness were evaluated at 25 °C using a UV-vis spectrophotometer (Lambda 25; PerkinElmer, Waltham, MA, USA), which is often used to evaluate the transparency of glassy plastics [5,6,25]. Light transmittance in the visible wavelength was measured as a function of the wavelength from 200 to 800 nm.

Light transmittance was also evaluated using an optical microscope (DMLP; Leica Microsystems, Wetzlar, Germany) equipped with a hot stage (FP90; Mettler-Toledo, Greifensee, Switzerland) to evaluate the LCST. One eyepiece was replaced with a photo-detector (PM16-121; Thorlabs, Newton, MA, USA) to determine the light intensity after passing through a color filter (633 nm) [26]. The sample sandwiched by cover glasses was heated at 10 °C/min from 200 °C.

The dependence on the temperature of the oscillatory tensile moduli in the solid state was investigated at 10 Hz using a dynamic mechanical analyzer (E-4000; UBM, Muko, Japan). The heating rate was 2 °C/min. The rectangular samples with a length of 15 mm and a width of 5 mm, cut out from the compression molded film, were employed for the measurement. The dependence on the angular frequency of the oscillatory shear moduli in the molten state was investigated at various temperatures using a cone-and-plate rheometer (AR2000ex; TA Instruments, New Castle, DE, USA). The diameter of the cone was 25 mm *φ*, and the cone angle was 4°.

Capillary extrusion [27] was carried out using a twin capillary rheometer (Rosand RH7; Netzsch, Selb, Germany) at various temperatures. A circular die (length: 16 mm; diameter: 1 mm) was used. The entrance angle of the die was 2π. The ambient temperature was maintained at 25 °C. The extruded strands were collected and cut perpendicular to the flow direction into circular sections (approximately 1 mm thick) by a razor blade. The circular sections and extruded strands were then examined using a stereomicroscope (S6E; Leica Microsystems) to determine their transparency.

## 3. Results and Discussion

### 3.1. Miscibility of the Blends

Figure 1 shows the light transmittance as a function of wavelength for PC/PMMA (70/30) films obtained by compression molding at various temperatures. Photographic images of two films processed at 230 and 250 °C are also shown. After taking surface reflection into consideration, which is typically 12% [25], the films processed at/below 230 °C were transparent. The pure PC film exhibited the same values (not shown here). In contrast, the blend film processed at 250 °C was opaque and had low light transmittance values, especially in the low wavelength region. This can be attributed to light scattering, which was obvious at long wavelengths. Owing to the huge difference between the refractive indices of PC and PMMA, there was intense light scattering when the sizes of the separated phases were within the visible wavelength. These results indicate that the LCST of PC/PMMA (70/30) is between 230 and 250 °C.

The LCST of PC/PMMA (80/20) was above 250 °C. Therefore, the compression-molded film prepared at 250 °C was transparent and had a similar light transmittance to the pure PC film (not presented here). Figure 2 shows the light intensity under an optical microscope as a function of temperature. As shown in the figure, the light transmittance decreased rapidly at approximately 270 °C, indicating the LCST of the sample.

Figure 3 shows the temperature dependencies of the tensile storage modulus (*E*′) and the loss modulus (*E*″) at 10 Hz for three sample films prepared by compression molding at 250 °C. However, the PC/PMMA (70/30) film was processed at 230 °C, i.e., below the LCST. The samples all demonstrated the typical viscoelastic properties of amorphous polymers. They all exhibited a sharp decrease in *E*′ and a single peak in the *E*″ curve, both of which can be attributed to a glass-to-rubber transition. The addition of PMMA shifted the peaks in the *E*″ curves to lower temperatures, demonstrating that both blend samples were miscible. The peak temperatures, i.e., *T_g_* values, were as follows: 160.9 °C for PC, 146.3 °C for PC/PMMA (80/20), and 140.0 °C for PC/PMMA (70/30). According to the Fox formula (Equation (1) [28]), the *T_g_* of the PMMA was estimated to be 97.5 °C:(1)$$\frac{1}{T_{g}\left(blend\right)}=\frac{w_{1}}{T_{g1}}+\frac{w_{2}}{T_{g2}}$$
where *w_i_* is the weight fraction and *T_gi_* is the *T_g_* of the *i*-th component.

This *T_g_* value was 10 °C lower than that of a conventional PMMA [29]. This was as expected because the PMMA sample used in the present study had a low molecular weight. The *E*″ values for the blends in the low temperature region were higher than those of pure PC, which became obvious for the PC/PMMA (70/30) film. This is ascribed to the β-dispersion of PMMA [30]. The E’ value in the glassy region was also enhanced by the PMMA addition, suggesting that PMMA acted as an antiplasticizer that could reduce thermal expansion and gas permeability [31].

Figure 4 shows the angular frequency dependencies of the shear storage modulus *G*′ and loss modulus *G*″ in the molten state. The reference temperature (*T_r_*) was 250 °C. The slopes of the *G*′ curves were 2 for both PC and PC/PMMA (80/20), and the slopes of the *G*″ curves were 1 for both PC and PC/PMMA (80/20), suggesting that PC/PMMA (80/20) was in the miscible state. This result corresponded with Figure 2 and Figure 3. The modulus decrease was obvious for PC/PMMA (70/30). This is reasonable because the content of the low modulus component increased. In the case of PC/PMMA (70/30), the *G*′ values in the low frequency region seemed to be slightly higher at 250 °C, although the data were scattered. Considering that the LCST of the blend was between 230 and 250 °C, this may be attributed to a long-time relaxation. When phase separation occurs, in which the continuous phase comprises a high-viscosity component, a relaxation attributed to interfacial tension is known to appear, which was predicted by the emulsion model [32,33]. For pure PC and PC/PMMA (80/20), the time–temperature superposition principle was applicable.

### 3.2. Capillary Extrusion

Capillary extrusion was performed using a circular die. The length *L* was 16 mm, and the diameter *D* was 1 mm. The shear stress *σ* and shear rate $\dot{\gamma }$ on the wall were calculated from the pressure drop Δ*P* and volume flow rate *Q* as follows [27]:(2)$$\sigma =\frac{D\;\Delta P}{4L}$$
(3)$$\dot{\gamma }=\frac{32Q}{\pi D^{3}}$$

Strictly speaking, the Bagley correction is required to evaluate the pressure drop in the die, i.e., Δ*P*/*L* [27]. However, Δ*P*/*L* must not be homogeneous in the die land due to the anomalous pressure increase near the die entry, especially at low temperatures. Furthermore, the Rabinowitsch correction, which is required for non-Newtonian fluids [27], was not performed, because the flow curve was quite different from those of conventional polymer melts, as shown later.

Figure 5 shows the flow curves, i.e., shear stress plotted against shear rate, of the PC/PMMA (70/30) film at various temperatures. It was impossible to measure the shear stress at high shear rates and at 200 °C because the pressure exceeded the limitations of the machine. It was found that the stress increase at a high shear rate was obvious at least at 200 °C, which is not seen in most polymer melts.

In general, the time–temperature superposition principle is applicable to the shear rate–shear stress curves of a simple polymer liquid. Therefore, the master curve was obtained for this blend by simple horizontal shifts. The results are shown in Figure 6, with dashed lines denoting the applied pressure. The reference temperature *T_r_* was 250 °C. The shift factor was the same as shift factors obtained by oscillatory measurements. Although most data were superposed onto each other, especially at low shear stresses, an upper deviation was detected in the high shear stress region at 200 °C and 210 °C. This deviation was obvious beyond 50 MPa as a pressure. Except for the upper deviated data, others were superposed onto each other including the data at 250 °C, i.e., beyond LCST. This result suggested that the phase separation did not greatly affect the shear stress for the blend system.

As mentioned in the introduction, the shear stress/viscosity becomes sensitive to applied pressure when the pressure *P* is beyond 10 MPa in general. This can be expressed by the following equation [9]:(4)η(P)=η(0)exp(αP)
where *η* is the shear viscosity and *α* is the coefficient. The stress/viscosity increase is attributed to the reduction of free volume. In many polymers, including polycarbonates, *α* is approximately 2.0 × 10^−8^ (Pa^−1^) [10]. According to Equation (4), the viscosity increases 20% at 10 MPa and 640% at 100 MPa. The results shown in Figure 5 and Figure 6 indicate that the stress increase was lower than that predicted by Equation (4). In fact, the time–temperature superposition principle was still applicable when the pressure was approximately 10 MPa. This must be attributed to the measurement method. During capillary extrusion, the pressure decreases as the material passes through the die. Therefore, high pressure was only applied at the die entrance, i.e., the top of the die. Furthermore, the shear stress was calculated assuming that the pressure gradient was constant throughout the whole die. Therefore, the actual shear stress in most of the die was lower than the values in the figures. However, the results demonstrated that high pressure was applied to the sample at least in the top area of the die. In this area, miscibility might be affected by the reduction of the free volume fraction. Under such high pressures, the interaction contribution plays an important role in the Flory–Huggins parameter [13].

The effect of the applied pressure on PC/PMMA (70/30) was obvious at 200 °C, as shown in Figure 7. In the figure, the top view of the cut strand as well as the side view of the strand are shown. The diameter of the strand was around 1 mm. It was found that the strand extruded at or below 43 s^−1^ was transparent. In contrast, the strand extruded at 93 s^−1^ was opaque throughout, with a gross melt fracture due to high elongational stress [34,35,36]. This pressure-induced demixing indicates that the interaction contribution to the Flory–Huggins parameter was positive for PC/PMMA, i.e., PC/PMMA was a repulsive system.

Figure 8 shows the PC/PMMA (70/30) strands extruded at 210 °C. The miscibility of the blend was significantly sensitive to the applied shear rate. As with the compression-molded film prepared at 210 °C, the strand was transparent under a low shear rate, e.g., 200 s^−1^. At 430 s^−1^, only the exterior of the strand became opaque, whereas the interior was transparent. Considering that the shear rate in the strand is a function of the distance from the center, the critical shear rate for shear-induced demixing was approximately 430 s^−1^ at this temperature. Furthermore, a strand with the opposite contrast was obtained at 930 s^−1^, demonstrating that shear-induced mixing occurs at high shear rates. Because the shear rate in the center of the strand was zero, the strand must have had three layers, i.e., miscible in the outer layer, immiscible in the middle layer, and miscible in the center. At high shear rates, e.g., 2000 s^−1^, all areas became transparent again. Considering the distribution of the shear rate, however, some areas with a shear rate of approximately 430 s^−1^ may have a phase-separated structure, although the effect of the residence time on the morphology change should also be considered. Moreover, the effect of the applied pressure on the miscibility was affected by temperature, because the pressure at 2000 s^−1^ and 210 °C was almost the same as that at 93 s^−1^ and 200 °C.

The strands extruded at 230 °C, i.e., below LCST, were transparent, irrespective of the shear rate applied in the experimental range, i.e., below 2000 s^−1^. Moreover, any flow instabilities, such as shark-skin failure and gross melt fracture, were not detected at even 2000 s^−1^. Therefore, processing at this temperature is preferable to ensure the transparency of a product.

Figure 9 shows PC/PMMA (70/30) strands extruded at 250 °C. Because 250 °C was above their LCST, the strand produced at a low shear rate was opaque, as with the compression-molded film. However, at 2000 s^−1^, the strand was transparent, indicating shear-induced mixing.

The results in Figure 7, Figure 8 and Figure 9 indicate that even in the region in which the time–temperature superposition principle is applicable, miscibility cannot be predicted by shear stress and pressure. The discrepancy between the extrusion temperature and the LCST must also affect miscibility.

Figure 10 shows flow curves with pictures of extruded strands of PC/PMMA (80/20). There was a rapid increase in the shear stress due to a reduction in the free volume fraction at 200 s^−1^ and 210 °C. Measurements were not obtained beyond this shear rate due to pressure limitations. At high shear stress, i.e., under high pressure (and consequently with a reduced free volume fraction), the strand became opaque throughout, suggesting pressure-induced demixing. The other strands shown in Figure 10, including those extruded at 230 °C, were transparent, suggesting that the appropriate processing window was enlarged from the viewpoint of transparency as the PMMA content was reduced. The flow instability, however, occurred easily due to high shear stress. In fact, the strand at 93 s^−1^ and at 210 °C showed shark-skin failure.

## 4. Conclusions

The effects of the pressure and shear rate on the miscibility of blends of PC and PMMA were studied using a pressure-driven capillary rheometer. Because the PMMA sample used had a low molecular weight, the blends containing 20 and 30 wt.% PMMA had a miscibility window at temperatures below the LCST. The LCST was 230–250 °C for PC/PMMA (70/30) and approximately 270 °C for PC/PMMA (80/20) without shear flow under atmospheric pressure. Under high pressure, however, pressure-induced demixing, i.e., phase separation, was detected in both blends, in which shear stress was greatly enhanced by the reduction of the free volume fraction. Furthermore, PC/PMMA (80/20) exhibited shear-induced mixing at relatively low shear rates, and shear-induced demixing at high shear rates. When shear-induced mixing/demixing occurred, either the exterior or the interior of the strand became opaque, whereas pressure-induced demixing produced opacity throughout the strand. Currently, polycarbonates are being injection-molded at high pressures in industries. Therefore, information about pressure-induced demixing and shear-induced mixing/demixing is very important.

## Figures and Tables

**Figure 1 materials-15-02783-f001:**
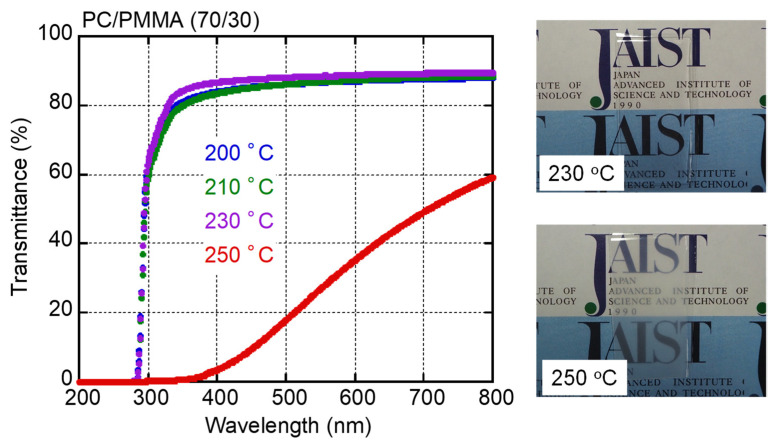
Light transmittance as a function of wavelength for compression-molded PC/PMMA (70/30) films prepared at various temperatures, with pictures of films processed at 230 and 250 °C. The films were approximately 0.5 mm thick.

**Figure 2 materials-15-02783-f002:**
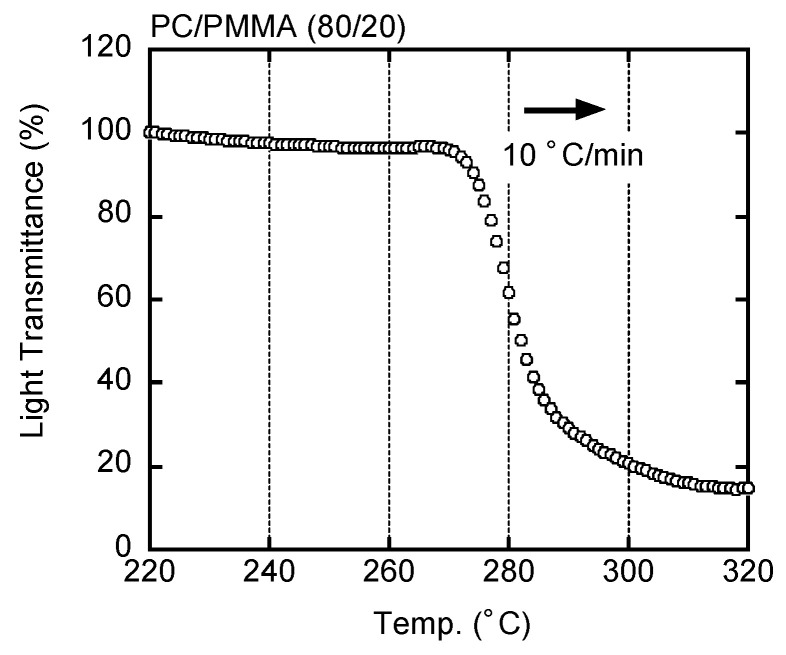
Light transmittance of PC/PMMA (80/20) during heating at 10 °C/min.

**Figure 3 materials-15-02783-f003:**
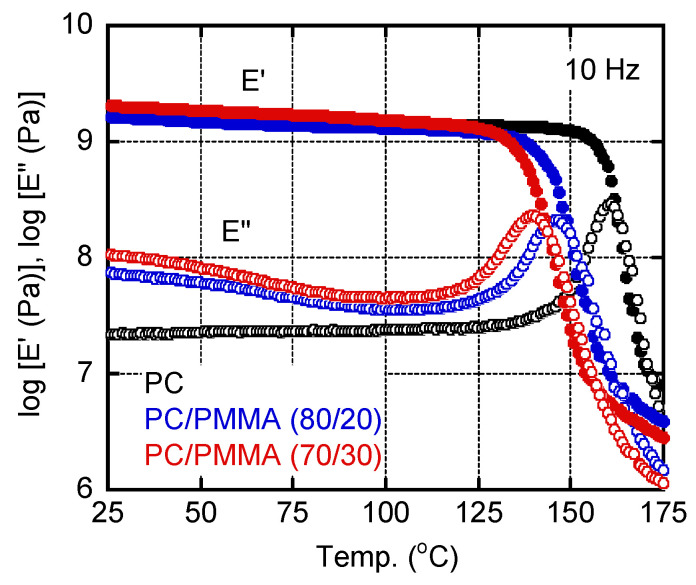
Temperature dependencies of the tensile storage modulus (*E*′) and the loss modulus (*E*″) at 10 Hz for PC, PC/PMMA (80/20), and PC/PMMA (70/30) films. The sample films were prepared by compression molding at 230 °C.

**Figure 4 materials-15-02783-f004:**
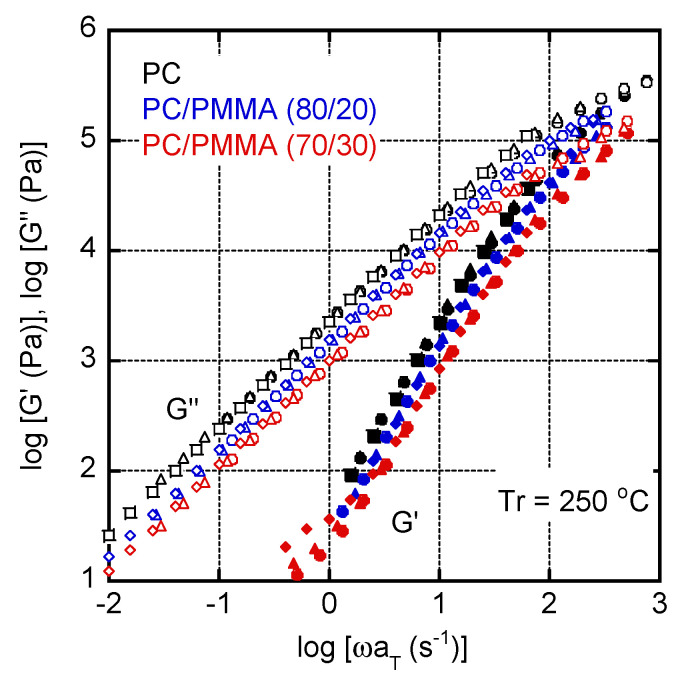
Master curves of shear storage modulus (*G*′) and loss modulus (*G*″) for PC, PC/PMMA (80/20), and PC/PMMA (70/30). The reference temperature *T_r_* was 250 °C. Circles represent data measured at 210 °C, triangles represent those at 230 °C, and diamonds represent those at 250 °C.

**Figure 5 materials-15-02783-f005:**
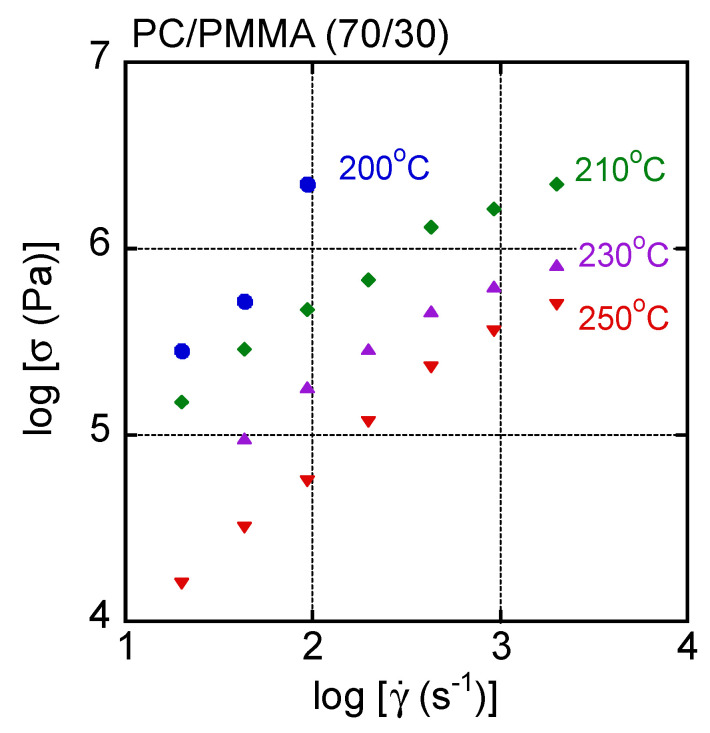
Flow curves of PC/PMMA (70/30) at various temperatures.

**Figure 6 materials-15-02783-f006:**
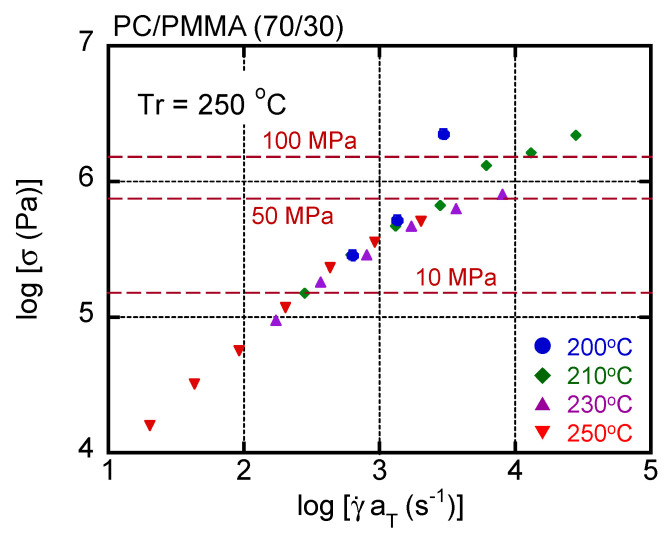
Master curve of shear stress as a function of shear rate in PC/PMMA (70/30). The reference temperature *T_r_* was 250 °C. Circles represent data measured at 200 °C, diamonds represent those at 210 °C, triangles represent those at 230 °C, and inverted triangles represent those at 250 °C.

**Figure 7 materials-15-02783-f007:**
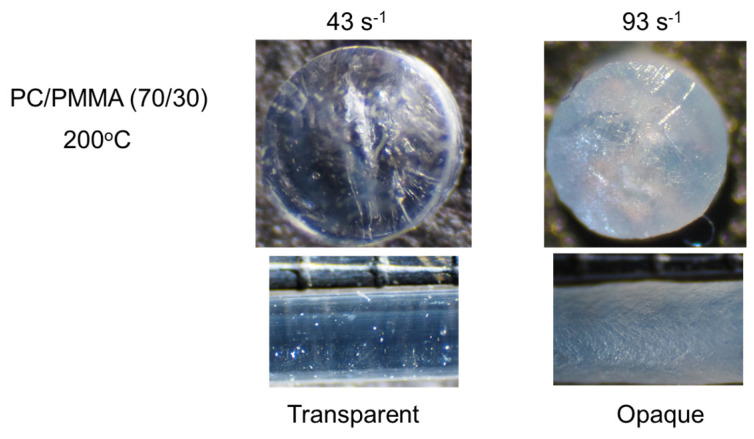
Strands of PC/PMMA (70/30) extruded at 200 °C. (**Top**) Top views of the cut strands, i.e., circular sections, and (**bottom**) side views of the strands.

**Figure 8 materials-15-02783-f008:**
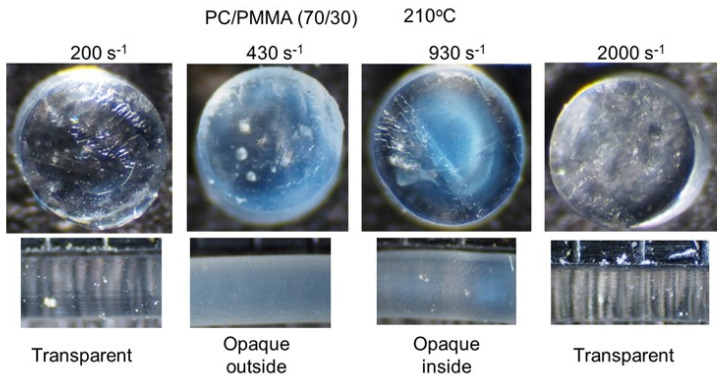
Strands of PC/PMMA (70/30) extruded at 210 °C. (**Top**) Top views of the cut strands, i.e., circular sections, and (**bottom**) side views of the strands.

**Figure 9 materials-15-02783-f009:**
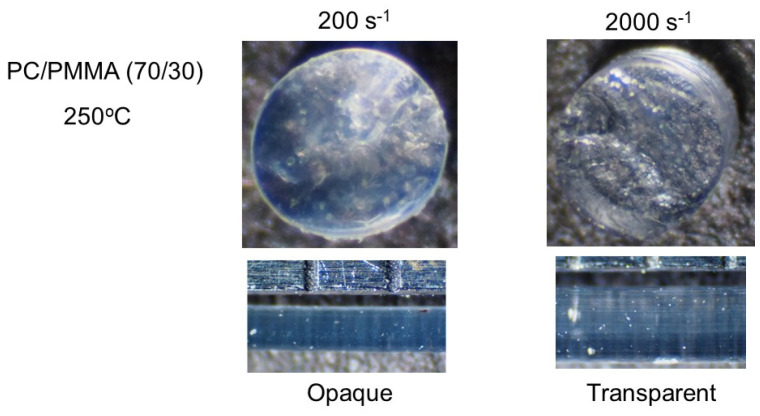
Strands of PC/PMMA (70/30) extruded at 250 °C. (**Top**) Top views of the cut strands, i.e., circular sections, and (**bottom**) side views of the strands.

**Figure 10 materials-15-02783-f010:**
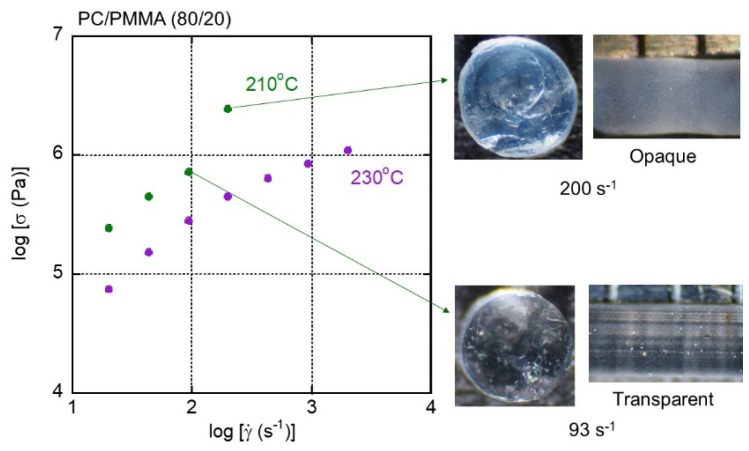
Flow curves and strands of PC/PMMA (80/20) extruded at 210 and 230 °C.

## Data Availability

Not applicable.

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
