# Peer review of "Complicated Structure Change during Capillary Extrusion of Binary Blends of Polycarbonate and Poly(methyl methacrylate)"

_materials, 2022, doi:10.3390/ma15082783_

Round 1

Reviewer 1 Report

In the paper, an effect of pressure and shear rate on the miscibility of PC/PMMA polyblend is studied using capillary rheometry.

The paper seems to be interesting, however, some explanations are necessary.

Polymeric materials are non-Newtonian fluids. The authors use the Newtonian solution to discuss their research results. Why ?

For non-Newtonian fluids the shear rate is different than for Newtonian fluids. Thus, the discussion of the results of investigations can be distorted. May be the discussion should be repeated and performed in relation to the non-Newtonian solutions.

Did the authors use the Rabinowitsch/Bagley correcctions ?

           Moreover:

  • literature is somewhat outdated,
  • shear stress is usually denoted by τ (tau),
  • shear stress is determined by pressure drop ΔP, and not by pressure P.

Author Response

In the paper, an effect of pressure and shear rate on the miscibility of PC/PMMA polyblend is studied using capillary rheometry. The paper seems to be interesting, however, some explanations are necessary.

  1. Polymeric materials are non-Newtonian fluids. The authors use the Newtonian solution to discuss their research results. Why ? For non-Newtonian fluids the shear rate is different than for Newtonian fluids. Thus, the discussion of the results of investigations can be distorted. May be the discussion should be repeated and performed in relation to the non-Newtonian solutions. Did the authors use the Rabinowitsch/Bagley correcctions ?

[response]

Yes, they are non-Newtonian fluids. Of course, as the reviewer suggested, the Rabinowitsch correction is often employed to correct the shear rate on the wall and the Bagley correction for pressure drop. However, as demonstrated in Figure 5, these simple corrections cannot be applicable due to the high pressure, leading to the increase in the stress. This stress increase occurs only at the top part of the die, and therefore, the pressure profile and velocity distribution are the function of the position in the die. Therefore, we used the characterization technique used for Newtonian fluids. We added the comments on this issue in the revised version.

[revised] Line 188

Strictly speaking, the Bagley correction is required to evaluate the pressure drop in the die, i.e., ΔP/L [27]. However, ΔP/L mustn’t be homogeneous in the die land due to the anomalous pressure increase near the die entry, especially at low temperatures. Furthermore, the Rabinowitsch correction, which is required for non-Newtonian fluids [27], was not performed, because the flow curve was quite different from those of conventional polymer melts as shown later.

Line 197

It was found that the stress increase at a high shear rate was obvious at least at 200 °C, which is not seen in most polymer melts.

  1. literature is somewhat outdated,

[response]

We added a couple of updated literatures.

  1. shear stress is usually denoted by τ (tau),

[response]

Yes, people working in the chemical engineering field prefer to use “tau”. However, some other researchers prefer to use “sigma”, because “tau” always represents relaxation time. Hope the reviewer understands it.

  1. shear stress is determined by pressure drop ΔP, and not by pressure P.

[response]

We use ΔP instead of P in the revised version.

Reviewer 2 Report

The manuscript titled "Complicated Structure Change During Capillary Extrusion of Binary Blends of Polycarbonate and Poly(methyl methacrylate)" is a short study on the effect of pressure and miscibility of PC/PMMA blends. It is an intersting, well-written manuscript that I recommend for publication in its current form.

Author Response

 Thank you so much for the positive comments and reviewing process.

Reviewer 3 Report

The experimental based investigations are always playing major role in the invention so the proposed work is greatly appreciable one. Some of the corrections are found out, which needs to be effectively solved by the authors.

  1. The proper literature surveys are needs to be given at sub sections 2.1, 2.2 and 2.3.
  2. More justification on the selection of instruments, environmental conditions, and details of test specimens are needed one under main section 2. 
  3. The relevant citations are missing at most of the analytical calculations. 
  4. Discussions on developed Figures are given in very less manner. Therefore it is strongly recommended to increase the discussions on all the Figures and their outcomes. 
  5. Please recheck and confirm the contents imposed in the articles. Especially, the units are needs to be checked. 
  6. It is strongly advised to create a separate section for nomenclatures. 

Author Response

The experimental based investigations are always playing major role in the invention so the proposed work is greatly appreciable one. Some of the corrections are found out, which needs to be effectively solved by the authors.

  1. The proper literature surveys are needs to be given at sub sections 2.1, 2.2 and 2.3.

[response]

Thank you so much for the positive comments and reviewing process. We added literatures at 2.1-2.3.

  1. More justification on the selection of instruments, environmental conditions, and details of test specimens are needed one under main section 2. 

[response]

We added the reason to use these instrument, environmental conditions, and details of test specimens.

  1. The relevant citations are missing at most of the analytical calculations.

[response]

We added literatures at 2.1-2.3.

  1. Discussions on developed Figures are given in very less manner. Therefore it is strongly recommended to increase the discussions on all the Figures and their outcomes.

[response]

We added more explanations on each figure.

[revised]

Line 152

The E” values for the blends in the low temperature region were higher than those of pure PC, which became obvious for the PC/PMMA (70/30) film. This is ascribed to the b-dispersion of PMMA [30]. The E’ value in the glassy region was also enhanced by the PMMA addition, suggesting that PMMA acted as an antiplasticizer that can reduces thermal expansion and gas permeability [31].

Line 167

The modulus decrease was obvious for PC/PMMA (70/30). This is reasonable because the content of the low modulus component increased.

Line 174

For pure PC and PC/PMMA (80/20), time-temperature superposition principle was applicable.

Line 197

It was found that the stress increase at a high shear rate was obvious at least at 200 °C, which is not seen in most polymer melts.

Line 209

This deviation was obvious beyond 50 MPa as a pressure. Except for the upper deviated data, others were superposed onto each other including the data at 250 °C, i.e., beyond LCST. This result suggested that the phase separation did not affect the shear stress greatly for the blend system.

Line 240

In the figure, the top view of the cut strand as well as the side view of the strand were shown. The diameter of the strand was around 1 mm. It was found that the strand extruded at or below 43 s-1 was transparent.

Line 273

The strands extruded at 230 °C, i.e., below LCST, were transparent, irrespective of the shear rate applied in the experimental range, i.e., below 2000 s-1. Moreover, any flow instabilities, such as shark-skin failure and gross melt fracture, were not detected even at 2000 s-1.

Line 296

The other strands shown in Figure 10, including those extruded at 230 °C, were transparent, suggesting that the appropriate processing window is enlarged from the viewpoint of transparency as the PMMA content is reduced. The flow instability, however, occurred easily due to high shear stress. In fact, the strand at 93 s-1 and at 210 °C showed shark-skin failure.

  1. Please recheck and confirm the contents imposed in the articles. Especially, the units are needs to be checked. 

[response]

Thank you. We checked all of them.

  1. It is strongly advised to create a separate section for nomenclatures. 

[response] We added the nomenclatures in the revised version.

Round 2

Reviewer 3 Report

The previously suggested comments are greatly discussed in the latest version.